# Parents’ and Guardians’ Willingness to Vaccinate Their Children against COVID-19: A Systematic Review and Meta-Analysis

**DOI:** 10.3390/vaccines10020179

**Published:** 2022-01-24

**Authors:** Feifan Chen, Yalin He, Yuan Shi

**Affiliations:** Department of Neonatology, Ministry of Education Key Laboratory of Child Development and Disorders, National Clinical Research Center for Child Health and Disorders, China International Science and Technology Cooperation Base of Child Development and Critical Disorders, Chongqing Key Laboratory of Pediatrics, Children’s Hospital of Chongqing Medical University, Chongqing 400014, China; 2017210134@stu.cqmu.edu.cn (F.C.); 2017221410@stu.cqmu.edu.cn (Y.H.)

**Keywords:** COVID-19, SARS-CoV-2, vaccine, vaccination willingness, vaccination hesitancy, determinants, predictors, systematic review

## Abstract

COVID-19 vaccination for children is crucial to achieve herd immunity. This is the first systematic review and meta-analysis to estimate parents’ and guardians’ willingness to vaccinate their children against COVID-19 and identify the determinants of vaccination intention. Systematic research was performed on the two databases (PubMed and EMBASE) from inception to 6 November 2021. Acceptance rates were pooled by use of a random-effects model and all predictors of vaccine acceptance were identified according to the health belief model (HBM) framework. This analysis was registered with PROSPERO (CRD42021292326) and reported in compliance with the PRISMA guidelines. Of 452 identified records, 29 eligible studies were included (N = 68,327 participants). The estimated worldwide vaccination acceptance rate was 61.40% (95% CI: 53.56–68.69%, I^2^ = 99.3%), ranging from 21.6% to 91.4% across countries and regions. In the determinant assessment, the age of parents and guardians, access to scientific information and recommendations, routine and influenza vaccination behavior, and the willingness of parents and guardians to vaccinate themselves were potentially significant predictors of the vaccination willingness. Given the limited quality and quantity of included articles, future studies with a rigorous design will be necessary for the confirmation of our findings.

## 1. Introduction

Declared as a global pandemic by the WHO on 11 March 2020, COVID-19 has spread to almost the whole world, resulting in serious disruptions in economics and society. Notably, over 200 million confirmed cases with over 5 million deaths have been recorded as of 30 November 2021 [1]. Various physical measures have been conducted to curb the spread of the virus, including mask-wearing and social distancing. As for children, closing or reducing time at school has been the primary approach [2]. However, social isolation has negative impacts on children’s mental health and well-being [3].

Vaccination is the key to mitigating the impact of COVID-19, which will enable children to return to normal activities [4]. A successful vaccination needs both adequate vaccine production as well as high levels of uptake. Inspiringly, over 100 different vaccine candidates have been developed since the genetic sequence for the virus was published in January 2020 [5]. According to statistical models, 60–72% of people need to be vaccinated to reach the threshold of herd immunity, in the case of vaccines that are 80% effective [6]. Considering the emergence of some new variants with high transmissibility, such as the recently discovered Omicron variant, a higher vaccination uptake may be needed [7].

While studies on the COVID-19 vaccine are ongoing, vaccine hesitancy, defined by the WHO as the delay in acceptance or refusal of vaccination despite the availability of vaccination services, might constitute an important obstacle to vaccination [8]. Parents and guardians are usually the decision makers on their children’s vaccination, and their vaccine hesitancy may result in surges of many vaccine-preventable diseases, such as measles [9]. Several studies have shown that the majority of measles outbreaks in the United States and European Union countries were associated with unvaccinated children [9,10,11,12]. During the COVID-19 pandemic, concerns about vaccine hesitancy have been further heightened. Therefore, to contain the pandemic, there is an urgent need to alleviate the COVID-19 vaccine hesitancy, which relies on a comprehensive understanding of its determinants. However, to our knowledge, there has been no systematic review and meta-analysis about COVID-19 vaccination willingness on children and its predictors yet.

Thus, this study aimed to (1) estimate parents’ and guardians’ willingness to vaccinate their children against COVID-19; (2) identify the predictors of vaccine willingness or vaccine hesitancy; and hoped to provide a reference for future vaccine coverage in children which would promote the development of herd immunity to end the pandemic.

## 2. Materials and Methods

This systematic review and meta-analysis were conducted in compliance with the Preferred Reporting Items for Systematic Reviews and Meta-Analyses (PRISMA) guidelines, and the protocol was available on PROSPERO (CRD42021292326).

### 2.1. Search Strategy

Systematic research was performed on the two databases (PubMed and EMBASE) from inception to 6 November 2021. Both controlled terms (e.g., MeSH terms in PubMed) and free-text terms were used based on the following four topics and their synonyms: children, COVID-19, vaccination, and survey. Table 1 categorized the search terms used in PubMed, and the search string was shown in Appendix A. Publications were limited to the English language, with no restrictions on the date, study type, or publication status.

### 2.2. Eligibility Criteria

The original records were selected based on the following inclusion criteria: (1) studies involved acceptance of COVID-19 vaccination for children; (2) studies with adult participants (>18 years of age) who were parents or guardians of minors; (3) studies provide specific survey data for pooling. Exclusion criteria were as follows: (1) non-primary documents: reviews, meta-analysis, editorials and other articles; (2) studies with subgroup-specific samples (e.g., healthcare workers or patients); (3) duplicate studies or databases.

Titles and abstracts were reviewed by two investigators (Chen and He) independently, and studies that satisfied the inclusion criteria were retrieved for full-text assessment. Disagreement was resolved by the third investigator (Shi).

### 2.3. Data Extraction

A data extraction form was constructed with Microsoft Excel. Two authors (Chen and He) independently extracted the following information from every included study: title, first author; survey characteristics including the survey type, method of contact, survey site, survey collection date, sampling method, main survey questions, and answer options; demographic data including the country, gender, age and level of education; results including vaccination willingness rates, sample size, determinants or predictors of vaccination willingness or hesitancy, reasons for vaccination acceptance or refusal and reported bias.

Vaccination willingness was defined as the proportion of participants willing or likely to vaccinate their children against COVID-19. Responses showing a positive attitude towards vaccination acceptance, or a negative tendency on vaccine refusal or hesitancy, were regarded as vaccination-willingness (Table 2).

All predictors of vaccine acceptance reported in the included studies were extracted according to the health belief model (HBM) framework [13]. This framework is derived from previous research in seasonal influenza vaccination behavioral determinants, which has been used to explain the factors related to immunization behaviors and can provide good support for complex and effective interventions [14]. Additionally, statistically significant predictors that are not defined by the HBM, such as socio-demographic factors and routine and influenza vaccination behavior, were also identified.

### 2.4. Quality Assessment

To critically appraise the methodological quality of included studies, an appraisal scale suggested by Iain Crombie was applied [15]. The scale contains the following seven indexes: (1) design is scientific; (2) data collection strategy is reasonable; (3) samples can represent the general population well; (4) sample response rate is reported; (5) the power of the test is reported; (6) the research purpose and method are reasonable; (7) the statistical method is reasonable. For every index, studies score 1 for “yes”, 0 for “no”, and 0.5 for “unclear”, respectively. The highest score on the scale is 7 points, lower than 4.0 is divided into C level, 4.0–5.5 is divided into B level, and 6.0–7.0 is divided into A level. Every included study was scored by two authors (Chen and He) independently, given the average points, and ranked as A or B or C according to the scale mentioned above.

### 2.5. Data Analysis

To estimate global COVID-19 vaccination willingness for children, a random-effect meta-analysis of single proportions was conducted.

All analyses were performed using R 4.1.2 software. After a logit transformation of the data, a random intercept logistic regression model was used for the meta-analyses. The within-study variation was estimated with the 95% confidence interval (CI) and the between-study variation was estimated with the maximum likelihood estimator for tau^2^. The Higgin’s and Thompsons’ I^2^ was used to assess heterogeneity. Subgroups analyses, meta-regression, and sensitivity analyses were conducted to explore the sources of heterogeneity. Publication bias was detected by Egger’s test. *p* < 0.05 was considered statistically significant.

Meanwhile, determinants associated with vaccination acceptance were systematically identified. The odds ratio (OR) was calculated and pooled when the data were appropriate and sufficient (e.g., age, gender). Reasons for vaccination willingness or refusal were identified as well, and the rates were pooled, respectively.

## 3. Results

### 3.1. Search Results and Study Characteristics

A total of 452 records were identified through the data search, and the titles and abstracts were screened for inclusion. In total, 97 potentially eligible articles were retrieved for full-text screening, and 29 were included in the meta-analysis (Figure 1). Included studies reported cross-sectional surveys from 16 countries and regions with a total of 68,327 participants. Near half of these studies were published in 2020 (*n* = 14) and the rest were in 2021 (*n* = 15). Populations in China were most frequently studied (*n* = 6), followed by the United States (*n* = 4) and Saudi Arabia (*n* = 4). The majority were conducted online (*n* = 20) and out of hospital (*n* = 18). Most studies were qualified as Level B (*n* = 18). An overview of the included studies was presented in Table 3.

### 3.2. Willingness on Children’s Vaccination

The estimated worldwide vaccination acceptance rate was 61.40% (95% CI: 53.56–68.69%, I^2^ = 99.3%) (Figure 2). The highest vaccination willingness of 91.4% was observed in the study of Bagateli in Brazil [16], and Marquez’s study in America reported the lowest vaccination intention (21.6%) [17].

### 3.3. Subgroup Analyses and Meta-Regression

Subgroup analyses were performed in Table 4, and it indicated that the willingness of COVID-19 vaccination for children varied across different continents (*p* < 0.0001), countries (*p* < 0.0001) and qualities of studies (*p* < 0.0001), contributing to the heterogeneity significantly. Of five continents, the majority of studies were conducted in Asia (14/29), with estimated vaccination willingness at 58.33% (95% CI: 47.96–68.01%). The highest willingness was observed in South America (91.42%, 95% CI: 88.63–93.57%), and Oceania reported the lowest vaccination acceptance (47.99%, 95% CI: 45.04–50.59%). Among different countries, Brazil (91.42%, 95% CI: 88.63–93.57%) and England (89.14%, 95% CI: 87.29–90.74%) had higher acceptance rates than other countries, and vaccination willingness in Canada (61.91%, 95% CI: 60.03–63.75%) and Korea (64.16%, 95% CI: 57.70–70.14%) were close to the global willingness. In addition, it was observed that vaccination acceptability was higher in Level A studies (74.23%, 95% CI: 65.61–81.31%), compared to studies of Level B (57.43%, 95% CI: 48.11–66.26%) and Level C (31.60%, 95% CI: 19.15–47.40%). However, subgroups of year, survey method and survey sites failed to explain the source of heterogeneity.

As shown in Table 5, the vaccination willingness for self significantly affected the heterogeneity of the results (*p* < 0.05) in both univariate and multivariate analyses, while survey sites, survey method, continent, the gender of parents and guardians, education level, age of parents and guardians, and routine and influenza vaccination behavior did not show a significant contribution to the heterogeneity.

### 3.4. Determinant Assessment

Determinants of vaccination willingness were statistically analyzed, and the results were demonstrated in Table 6. The determinants were divided into the following five main categories: (1) socio-demographic, (2) perceived susceptibility and severity, (3) perceived benefits and risks of vaccination, (4) cues of action, (5) attitude and beliefs, and routine and influenza vaccination behavior.

#### 3.4.1. Socio-Demographics

Education level (15/29), gender (12/29), and age of parents and guardians (8/29) were the most frequent predictive factors reported in the included studies.

There was contradictory evidence on whether educational level (OR: 1.3795, 95% CI: 0.8007, 2.3765) or gender of parents and guardians (OR: 1.1130, 95% CI: 0.5903, 2.0986) was an effective predictor. In nine studies [16,18,19,20,21,22,23,24,25], participants with a high school or lower degree were less likely to accept COVID-19 vaccination for their children compared to those with college or higher education, while the opposite results were found in the other six studies [26,27,28,29,30,31]. Compared to women, men were more likely to accept the vaccination in ten studies [22,24,25,27,28,29,32,33,34,35]. However, women’s acceptance rate was higher in the studies of Babicki and Wan [36,37].

The age of parents and guardians was possibly a significant predictor associated with vaccination willingness [16,21,22,26,28,36,38,39]. Parents and guardians aged < 30 were less likely to vaccinate their children against COVID-19 compared to those aged > 30 (OR: 0.6450, 95% CI: 0.4531–0.9180). The same trend was also observed in the association between children’s age and vaccination acceptance in five studies [22,26,33,38,40]. In addition, lower household income was found to be a negative factor of vaccination willingness in five studies [18,21,27,39,41]

#### 3.4.2. Perceived Susceptibility and Severity

The association between perceived susceptibility and severity of COVID-19 and vaccination acceptance was reported in nine studies. Eight studies found a positive relationship between perceived infection risk and severity and vaccine willingness [29,30,33,34,36,37,39,41]. Only one study showed the perception of COVID-19 disease risk and severity was not a significant factor in accepting the vaccination [42].

#### 3.4.3. Perceived Benefits and Risks of Vaccination

(Reasons for vaccination willingness or unwillingness)

Fifteen studies explored reasons for participants’ reluctance to vaccination. Of all the reasons, the safety of vaccines was the most frequently reported one, which almost 60.99% (95% CI: 48.37–72.30%) of the participants were concerned with. The second leading cause was the novelty or lack of evidence of vaccines, which accounted for about 54.40% (95% CI: 32.51–74.72%) of the respondents. Additionally, 24.20% (95% CI: 13.61–39.29%) doubted the effectiveness of vaccines. Meanwhile, eleven studies explored reasons for the acceptance of vaccination and the primary reason was to protect children and people around (55.31%, 95% CI: 41.39–68.44%).

#### 3.4.4. Attitude and Beliefs, and Routine and Influenza Vaccination Behavior

Of the included 29 studies, 13 articles indicated that parents’ or guardians’ intention to receive a COVID-19 vaccine for themselves was a significant independent factor associated with vaccination intention for their children, which was the most frequently reported predictor in our analysis. Eleven studies showed that participants who intended to vaccinate themselves were possibly more likely to accept the COVID-19 vaccination for their children [25,26,27,30,31,34,36,39,42,43,44], and the other two found adult vaccine hesitancy was negatively associated with vaccination willingness for children [19,40].

Routine and influenza vaccination behavior might be another positive predictor. Seven studies reported a positive relationship between routine and influenza vaccination behavior and COVID-19 vaccination willingness for children [18,19,32,33,38,39,40], while in the studies of Humble and Marquez [17,42], children’s routine vaccination status was not a significant determinant. Comprehensive analysis of nine studies mentioned above, parents and guardians who had given their children routine vaccines or seasonal influenza vaccines had a higher willingness rate than those who had not (OR: 2.1517, 95% CI: 1.2181, 2.7696).

#### 3.4.5. Cues to Action (Access to Scientific Information or Recommendations)

Nine studies stressed the importance of the access to scientific information or recommendations from public health authorities and physicians [17,19,21,24,26,28,30,33,36]. Participants exposed to scientific and positive information related to COVID-19 vaccines were more willing to have their children vaccinated. Three articles also indicated that lack of official information or misinformation was one of the main reasons for vaccine hesitancy among guardians [18,22,44]. Global misinformation spread through social media during the pandemic may pose challenges for future COVID-19 vaccination programs [30]. However, in the study of Altulaihi, the influence of social media was not a strong factor related to the parental acceptability of COVID-19 vaccination [38].

### 3.5. Publication Bias

A funnel plot and Egger’s test were performed to assess the publication bias and the results did not show evidence of publication bias (*p* > 0.05) (Appendix A).

## 4. Discussion

This is the first systematic review and meta-analysis of parents’ or guardians’ willingness to vaccinate their children against COVID-19. The worldwide vaccination acceptance rate was estimated at 61.40%, ranging from 21.6% to 91.4% across countries and regions. In the determinant assessment, the age of parents and guardians, access to scientific information and recommendations, routine and influenza vaccination behavior, and willingness of parents and guardians to vaccinate themselves were observed to be potentially significant predictors of the vaccination acceptance.

Children contribute significantly to the spread of COVID-19 as with other respiratory viruses [30]. Even if the pandemic slows down or even stops spreading locally, it is likely to regenerate and spread without adequate immunity. Indeed, this can be seen as recurring waves of the pandemic. However, if herd immunity can be achieved, the spread of COVID-19 may be halted. Effective herd immunity will require pediatric vaccination [18,45,46]. The vaccination of children has both direct benefits (protecting children from rare but severe cases of pediatric COVID-19 disease) and indirect benefits (protecting others by reducing the spread of the virus) [47], which has been proved to be successful in preventing many infectious diseases in whose transmission children play an essential role [48]. Therefore, improving the uptake of COVID-19 vaccination on children is vital to epidemic control.

By modifying the HBM framework, the determinants of vaccination willingness were comprehensively analyzed in our study. Among socio-demographic factors, the age of parents and guardians was an effective predictor. All the included eight studies demonstrated that parents and guardians aged < 30 were less likely to vaccinate their children against COVID-19 than those aged > 30. The reason may be that older parents and guardians were more experienced in information selection, and they could judge the pros and cons of vaccines more accurately. The children of younger parents and guardians tend to be younger and more vulnerable to the impact of vaccines, which may lead to vaccination hesitancy. The education level and gender of parents and guardians were not conclusive predictors. Contradictory results were reported on the association between high-level education and vaccination acceptance. Higher education gave parents and guardians access to more and better information about the virus and vaccine, providing them with more tools for decision making to avoid falling prey to conspiracy theories [22]. However, the fact that it had taken such an unprecedentedly short time to develop the vaccines also made people feel doubtful about its safety and effectiveness. Several included studies [24,25,27,35] showed that fathers were more willing than mothers to vaccinate their children against COVID-19, which was consistent with vaccine intention for other diseases, as a systematic review showed that women were less likely to be vaccinated during the 2009 global influenza pandemic [49]. It may be because men engage in riskier behaviors than do women from the psychological aspect [39]. However, mothers spend more time with their children than fathers and are more concerned about their children’s health-related illnesses in daily life, so women are considered the main decision makers regarding their children’s health [37]. Additionally, lower household income was observed to be a negative factor related to vaccination intention, which may hinder the availability of the COVID-19 vaccine [21,39]. When promoting vaccination, we should also consider economic factors.

In line with the expectation, parents and guardians who were worried about themselves and their children contracting COVID-19 were more willing to vaccinate their children than those without such worries. Similarly, parents were more likely to have their children vaccinated if the mortality rate in children increased following a mutation of the virus. However, most infected children are asymptomatic or have only mild symptoms [50,51]. The mild or asymptomatic course of COVID-19 among children may cause parents to feel less anxious and result in a low vaccination willingness [30]. However, it is worth noting that children, frequently referred to as asymptomatic carriers, may be responsible for the spread of the virus, representing an essential chain of viral transmission [52]. Therefore, high coverage of COVID-19 vaccination among children is necessary.

Vaccination is believed to be the most effective way to eliminate severe infectious diseases, and protection for children and people around was the main reason why parents and guardians intended to vaccinate their children in our analysis. However, school lockdown and online courses have been applied in many countries, which theoretically have reduced the possibility of children being infected, but also may weaken parents’ perception of the necessity to vaccinate their children. As for the vaccination hesitancy, the safety of vaccines was the most frequently reported contributor. Specifically, the novelty and side effects of COVID-19 vaccines were the two main concerning issues. During the pandemic, some vaccine candidates have been granted fast-track licensure by the US Food and Drug Administration [32], which may result in vaccination hesitancy. A considerable number of vaccine-hesitant parents cited mistrust with the rushed nature of testing in Hetherington’s study [21]. Parents opposed the use of COVID-19 vaccines in children without prior testing, and they were not willing to let their children take part in a COVID-19 vaccine or treatment trial [23]. As such, it is crucial for the government and medical professionals to communicate effectively with parents about novel vaccine types. Parents were also worried about potential side effects both instant and long-term [27]. Clarifying the side effects of COVID-19 vaccines may significantly reduce vaccination hesitancy.

Parents’ or guardians’ willingness to vaccinate themselves was the most significant predictor in our results. The more willing they were to vaccinate themselves, the more likely to vaccinate their children. Acceptance of vaccinating self may be a manifestation of their confidence and trust in COVID-19 vaccines. Several studies also reported that participants were more reluctant to vaccinate their children, compared to vaccinating themselves [19,20,31,41,44]. Parents and guardians are the decision makers for children’s vaccination. Aiming to improve COVID-19 vaccination uptake in children, strategies may need to be targeted on reducing vaccine hesitancy of their parents and improving their uptake firstly. Routine and influenza vaccination behavior was also a strong determinant. Several previous research has shown that parents often make the decision on their children’s vaccination of a newly developed vaccine based on attitudes towards established vaccines [42,53]. Our study showed that the children who were up-to-date on their immunization schedule or had a history of taking the seasonal influenza vaccine would be more likely to be vaccinated against COVID-19 by their parents and guardians. This was also observed during the H1N1 pandemic, in which those who had previously received the influenza vaccine were 21 times more likely to receive the H1N1 vaccine [54]. Moreover, Humble’s study suggested that parents’ intentions about COVID-19 vaccination are better predicted by previous decisions regarding influenza vaccination than routine childhood vaccination [21]. This may be due to parental concerns with the effectiveness or necessity of the influenza vaccine in comparison to routine childhood vaccines, given the historically low rates of influenza uptake in children [31].

Scientific information and recommendations were observed to be important in improving the vaccination rate, consistent with some findings of previous vaccine acceptance studies [13,14,55,56]. As we know, there has been a lot of misinformation, conspiracy theories as well as blatant anti-vaccine propaganda during the pandemic [57]. What is worse, previous research has shown that individuals are more likely to absorb negative information than positive information during a disease outbreak [58]. A considerable number of parents claimed that the virus was a biological weapon intended to manipulate human genetic material through vaccination and that the threat posed by COVID-19 was exaggerated [29]. Social media, a powerful tool for disseminating information, was also a platform for false data, unverified rumors, and even malicious misinformation [30]. The increasing influence of social media and explosion of the available information in recent years made it difficult for people to distinguish between true and false information. Parents uncertain about vaccinating their children against COVID-19 were waiting for official advice. The lack of information offered by the government about the COVID-19 vaccine appeared to create a vacuum, with parents eager for advice and therefore turning to other sources to fill this informational void, which sometimes resulted in potential misinformation [44]. Public health officials at federal, state, and local levels and primary care professionals were the most trusted sources of information about COVID-19 vaccines [24]. Only when information was obtained from a medical doctor or scientific reports was the highest percentage of parents willing to get their child vaccinated as soon as possible [36]. Therefore, it is suggested that public health officials engage in vaccination community campaigns and take advantage of the media to raise people’s awareness, to inform the importance of vaccines, and provide scientific information and recommendations for parents and guardians on vaccinating their children against COVID-19. It also should be noted that too aggressive messages and campaigns often have an opposite effect and arouse fear in the recipients, which may lead to their ignoring or repressing on the information obtained [36]. It may be explained by a psychological theory that people who feel too much compulsion to perform an activity feel like behaving in the complete opposite way [36].

When interpreting our findings, several limitations of this review should be considered. Firstly, this review is limited to English publications, and publications in other languages are not included. Secondly, the representativeness of the sample was uncertain in most studies. The majority of included studies relied on web-based questionnaires/surveys and used convenience sampling and snowball sampling methods when recruiting participants. The accessibility of the Internet (completing an online survey) may be a barrier for people living in a poor condition and lack of a random sampling process may result in unrepresentativeness. Data were self-reported and subject to response bias, which may have led to the predominance of female respondents in these studies. Selection bias could not be ruled out either, because several surveys were conducted in vaccination clinics, emergency departments, or other medical settings. Thirdly, because of the data insufficiency, other determinants that might contribute to vaccination acceptance were not investigated in detail, including occupation, household income, COVID-19 infection history in the household, chronic diseases or immune-compromised conditions, prior adverse or side effects of vaccines, and so on. Additionally, all the included studies were limited by the dynamic nature of the COVID-19 pandemic; the acceptance of the COVID-19 vaccine is dynamic and changes with legislation and public awareness policies. Many reviewed studies proposed a hypothetical vaccine, and once available and tested, participants may learn new information that may change their minds with regard to vaccinating their children. The reported vaccination willingness may not reflect actual vaccination behavior as well. Therefore, new research, especially longitudinal studies at different intervals, will be needed to investigate the actual vaccination behavior and attitudes when vaccination programs for children commence. Finally, the heterogeneity of this analysis remains at a high level, even though subgroup and meta-regression analyses were performed. The difference in vaccination acceptance ranged greatly among countries and regions. Several previous studies also suggested significant geographic and time-dependent variability in willingness to vaccinate children and the multitude of factors related to vaccination acceptance [18,36,39,41]. All of these imply substantial differences between populations. Therefore, the generalizability of our results is limited and strategies focusing on improving the vaccine uptake should be in line with determinants that are relevant within their specific setting and population.

In addition to these determinants observed in this study, uninvestigated factors mentioned above, scientific random sampling, and focusing on a specific group of people instead of the general population may be good suggestions for future studies. Particular attention to context-specific conditions also matters.

## 5. Conclusions

In general, parents and guardians around the world had moderate acceptance of children’s COVID-19 vaccination (61.40%). The age of parents and guardians, access to scientific information and recommendations, routine and influenza vaccination behavior, and the willingness of parents and guardians to vaccinate themselves were observed to be potentially significant predictors of vaccination willingness. However, given the limited quality and quantity of included articles, future studies with a rigorous design will be necessary for the confirmation of our findings.

## Figures and Tables

**Figure 1 vaccines-10-00179-f001:**
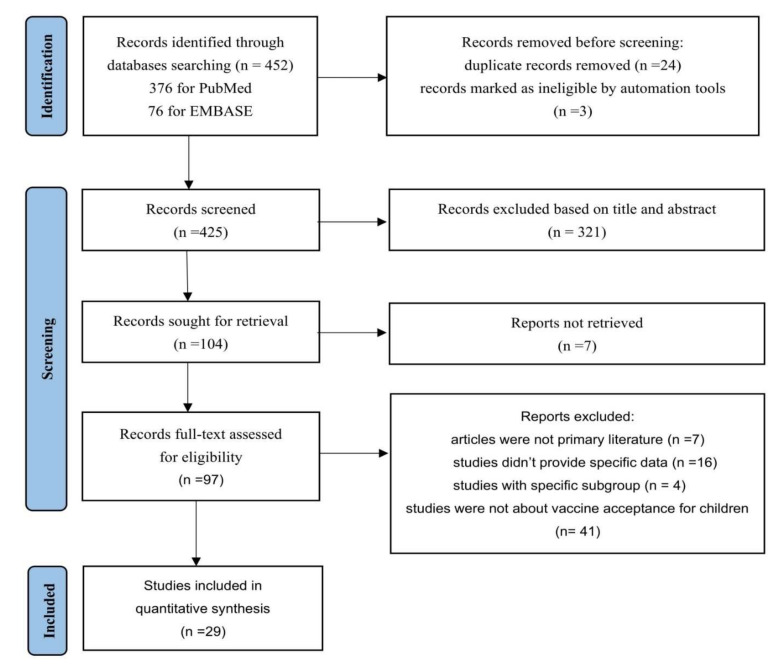
PRISMA 2020 flow diagram for identification of studies via databases.

**Figure 2 vaccines-10-00179-f002:**
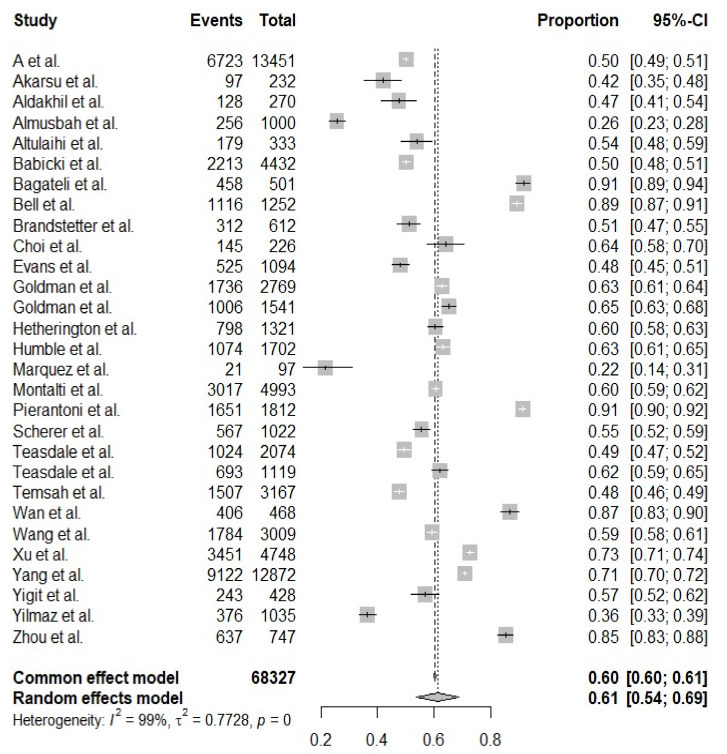
The forest plot of worldwide vaccination acceptance rate.

**Table 1 vaccines-10-00179-t001:** Search terms used in PubMed.

Mesh	Synonyms [Title/Abstract]
Infant OR Child OR Adolescent	Children OR adolescents OR adolescence OR teen OR teens OR teenager OR teenagers OR youth OR youths
SARS-CoV-2 OR COVID-19	Coronavirus Disease 2019 Virus OR 2019 Novel Coronavirus OR 2019 Novel Coronaviruses OR SARS-CoV-2 Virus OR SARS-CoV-2 Virus OR SARS-CoV-2 Viruses OR 2019-nCoV OR COVID-19 Virus OR COVID 19 Virus OR COVID-19 Viruses OR SARS Coronavirus 2 OR Severe Acute Respiratory Syndrome Coronavirus 2 OR COVID 19 OR COVID-19 Virus Disease OR COVID 19 Virus Disease OR COVID-19 Virus Diseases OR COVID-19 Virus Infection OR COVID 19 Virus Infection OR COVID-19 Virus Infections OR 2019-nCoV Infection OR 2019 nCoV Infection OR 2019-nCoV Infections OR Coronavirus Disease-19 OR Coronavirus Disease 19 OR 2019 Novel Coronavirus Disease OR 2019 Novel Coronavirus Infection OR 2019-nCoV Disease OR 2019 nCoV Disease OR 2019-nCoV Diseases OR Coronavirus Disease 2019 OR SARS Coronavirus 2 Infection OR SARS-CoV-2 Infection OR SARS-CoV-2 Infection OR SARS-CoV-2 Infections
COVID-19 Vaccines OR Vaccines OR Vaccination	Vaccine OR vaccine OR vaccins OR vaccined OR vaccinal OR vaccinate OR vaccinated OR vaccinates OR vaccinator OR vaccinators OR vaccinating OR Vaccinations OR Active Immunization OR Active Immunizations
Surveys and Questionnaires OR Health Surveys	Survey OR surveys OR surveyed OR surveying OR questionnaire OR questionnaires OR questionnair

**Table 2 vaccines-10-00179-t002:** Survey question responses categorized as vaccination-willingness.

Attitude	Response
Vaccination willingness	Yes/Yes, definitely/Yes, as soon as it will be possible/Very likely Agree/Likely/Fairly likely/Somewhat likely Yes, but only in a few months (up to a year)/Yes, but in more than a year/Unsure but leaning towards yes
Vaccination refusal or hesitancy	No/No, never/No, definitely not/Not at all likely Very unlikely/Quite unlikely Unlikely/Somewhat unlikely/Not likely/No, but maybe I will consider it in the future/Unsure but leaning towards no Neutral/Undecided/Uncertain/Not sure/I cannot decide/I don’t know

**Table 3 vaccines-10-00179-t003:** Characteristics of included studies.

Author	Year	Continent	Country	Survey Sites	Survey Method	Vaccination Willing (n)	Sample Size (n)	Quality of Study
Bagateli et al. [16]	2021	South America	Brazil	hospital	offline	458	501	B
Marquez et al. [17]	2020	North America	the United States	hospital	offline	21	97	C
Akarsu et al. [18]	2020	Asia	Turkey	non-hospital	online	97	232	C
Aldakhil et al. [19]	2021	Asia	Saudi Arabia	hospital	offline	128	270	B
Brandstetter et al. [20]	2020	Europe	Germany	non-hospital	online	312	612	B
Hetherington et al. [21]	2020	North America	Canada	hospital	online	798	1321	B
Montalti et al. [22]	2021	Europe	Italy	non-hospital	online	3017	4993	A
Pierantoni et al. [23]	2020	Europe	Italy	non-hospital	online	1651	1812	B
Scherer et al. [24]	2021	North America	the United States	non-hospital	online	567	1022	B
Teasdale et al. [25]	2021	North America	the United States	non-hospital	online	1024	2074	B
Temsah et al. [26]	2021	Asia	Saudi Arabia	non-hospital	online	1507	3167	B
Wang et al. [27]	2020	Asia	China	non-hospital	offline	1784	3009	A
Yang et al. [28]	2020	Asia	China	non-hospital	online	9122	12872	A
Yigit et al. [29]	2020	Asia	Turkey	hospital	offline	243	428	B
Yilmaz et al. [30]	2021	Asia	Turkey	non-hospital	online	376	1035	B
Zhou et al. [31]	2020	Asia	China	hospital	offline	637	747	A
Almusbah et al. [32]	2021	Asia	Saudi Arabia	non-hospital	online	256	1000	B
Goldman et al. [33]	2020	multi-continent	multi-country	hospital	online	1006	1541	A
Teasdale et al. [34]	2021	North America	the United States	non-hospital	online	693	1119	B
Xu et al. [35]	2020	Asia	China	non-hospital	online	3451	4748	A
Babicki et al. [36]	2021	Europe	Poland	non-hospital	online	2213	4432	B
Wan et al. [37]	2021	Asia	China	non-hospital	offline	406	468	A
Altulaihi et al. [38]	2021	Asia	Saudi Arabia	hospital	offline	179	333	B
Goldman et al. [39]	2021	multi-continent	multi-country	hospital	online	1736	2769	B
A et al. [40]	2020	Asia	China	hospital	online	6723	13451	B
Bell et al. [41]	2020	Europe	England	non-hospital	online	1116	1252	A
Humble et al. [42]	2020	North America	Canada	non-hospital	online	1074	1702	A
Choi et al. [43]	2021	Asia	Korea	hospital	offline	145	226	B
Evans et al. [44]	2021	Oceania	Australian	non-hospital	online	525	1094	B

**Table 4 vaccines-10-00179-t004:** The results of subgroup analyses by characteristics of population.

Subgroups	No. of Studies	Acceptance Rate (%) (95% CI)	I^2^ (%)	*p*-Value
Continent				<0.0001
Asia	14	58.33 [47.96, 68.01]	99.5	
Europe	5	72.66 [51.99, 86.71]	99.6	
South America	1	91.42 [88.63, 93.57]	-	
Oceania	1	47.99 [45.04, 50.95]	-	
Multi-continents	2	63.72 [61.90, 65.51]	65.1	
North America	6	52.25 [40.77, 63.51]	96.2	
Country				<0.0001
China	6	72.57 [60.66, 81.95]	99.7	
Turkey	3	44.78 [35.09, 54.89]	96.1	
Saudi Arabia	4	42.95 [32.21, 54.40]	98.1	
Poland	1	49.93 [48.46, 51.40]	-	
Brazil	1	91.42 [88.63 93.57]	-	
England	1	89.14 [87.29, 90.74]	-	
Germany	1	50.98 [47.02, 54.93]	-	
Korea	1	64.16 [57.70, 70.14]	-	
Australian	1	47.99 [45.04, 50.95]	-	
Multi-country	2	63.72 [61.90, 65.51]	65.1	
Canada	2	61.91 [60.03, 63.75]	56.3	
the United States	4	47.03 [32.02, 62.59]	96.3	
Italy	2	79.80 [51.30, 93.68]	99.8	
Year				0.3184
2020	14	65.26 [53.74, 75.24]	99.5	
2021	15	57.63 [47.34, 67.30]	98.7	
Survey method				0.4468
online	20	59.17 [50.80, 67.04]	99.5	
offline	9	66.21 [49.33, 79.77]	98.4	
Survey sites				0.9061
hospital	11	61.99 [49.04, 73.43]	98.7	
non-hospital	18	61.04 [51.08, 70.16]	99.5	
Quality of study				< 0.0001
A B	9 18	74.23 [65.61, 81.31] 57.43 [48.11, 66.26]	98.9 98.9	
C	2	31.60 [19.15, 47.40]	91.4	

**Table 5 vaccines-10-00179-t005:** The results of univariate and multivariate analyses on heterogeneity.

Variables		Coefficient	95% CI	Std. Err	*p*-Value
Univariate Analysis					
Survey sites	Hospital	ref	ref	ref	ref
	Non-hospital	−0.0401	(−0.7038, 0.6236)	0.3386	0.9057
Survey method	Offline	ref	ref	ref	ref
	Online	−0.3043	(−0.9927, 0.3841)	0.3512	0.3863
Continent	America	ref	ref	ref	ref
	Asia	−0.0607	(−0.8293, 0.7079)	0.3922	0.8770
	Europe	0.5792	(−0.3907, 1.5492)	0.4949	0.2418
	Multi-continent	0.1782	(−1.1466, 1.5030)	0.6759	0.7921
	Oceania	−0.4779	(−2.2446, 1.2887)	0.9014	0.5960
Gender of parents and guardians	Male	ref	ref	ref	ref
	Female	0.7219	(−1.5641, 3.0079)	1.1664	0.5360
Education level	High school or lower	ref	ref	ref	ref
	College or higher	−0.2851	(−2.1543, 1.5841)	0.9537	0.7650
Age of parents and guardians	≥30	ref	ref	ref	ref
	<30	0.3633	(−4.4087, 5.1354)	2.4348	0.8814
Routine and influenza vaccination behavior	No	ref	ref	ref	ref
	Yes	−0.4782	(−2.5808, 1.6244)	1.0728	0.6558
Vaccination willingness for self	No	ref	ref	ref	ref
	Yes	3.8195	(2.4483, 5.1906)	0.6996	<0.001
**Multivariate Analysis**					
Gender of parents and guardians	Male	ref	ref	ref	ref
	Female	1.0154	(−1.6611, 3.6919)	1.3656	0.4571
Education level	High school or lower	ref	ref	ref	ref
	College or higher	1.3974	(−0.7758, 3.5706)	1.1088	0.2076
Vaccination willingness for self	No	ref	ref	ref	ref
	Yes	3.7930	(2.0050, 5.5811)	0.9123	<0.001

**Table 6 vaccines-10-00179-t006:** The results of pooled rates and pooled OR of studied determinants.

Studied Items	No. of Studies	Pooled Rates (%) (95% CI)		Heterogeneity (I^2^, %)	Test of Heterogeneity (*p*-Value)
Reasons for Unwillingness to Vaccinate the Children					
Safety, Side effect [17,18,21,25,26,27,28,30,31,32,33,35,36,38,41]	15	60.99 (48.37, 72.30)		99.4	<0.01
Novelty, Lack of evidence [18,21,26,28,30,32,33,35,36,38,41]	11	54.40 (32.51, 74.72)		99.6	<0.01
Effectiveness [18,21,26,28,30,31,32,33,35,36,38,41]	12	24.20 (13.61, 39.29)		98.8	<0.01
Reasons for willingness to vaccinate the children					
Protection for children and people around [17,18,24,26,27,28,30,31,33,41,42]	11	55.31 (41.39, 68.44)		99.1	<0.01
**Studied Determinants**	**No. of Studies**	**Pooled OR** **(95% CI)**		**Heterogeneity** **(I^2^, %)**	**Test of Effect** **(*p* Value)**
Gender of parents and guardians [16,18,25,28,30,31,32,33,34,36,37]	11	Female	ref	ref	ref
		Male	1.1130 (0.5903, 2.0986)	98.8	0.7406
Education level [16,18,25,28,30,33,34,36,37]	9	College or higher	ref	ref	ref
		High school or lower	1.3795 (0.8007, 2.3765)	93.6	0.2464
Age of parents and guardians [16,18,25,28,30,34,36,37]	8	≥30	ref	ref	ref
		<30	0.6450 (0.4531, 0.9180)	91.6	<0.05
Routine and influenza vaccination behavior [18,21,30,31,32,33,36,37,39]	9	No	ref	ref	ref
		Yes	2.1517 (1.2181,2.7696)	92.4	<0.01

## Data Availability

Data can be requested from the author via: chenfeifan2021@163.com.

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
