# Peer review of "Parents’ and Guardians’ Willingness to Vaccinate Their Children against COVID-19: A Systematic Review and Meta-Analysis"

_vaccines, 2022, doi:10.3390/vaccines10020179_

Round 1
Reviewer 1 Report
This paper, Parents’ and Guardians’ Willingness to Vaccinate their Children against COVID-19: A Systematic Review and Meta-Analysis, provides info on vax. The paper considers variations in vax hesitancy by region of the world which is helpful.
It would be helpful to understand which studies provided data for the modeling, not just number of studies. Also, adding region into the models might also be helpful. One finding is that those who get sick with COVID, potentially the hospitalized patients, have a different view on vax than non-hospitalized. Adding this variable into the model might also be helpful. At least providing the results of whether or not this was important in Table 5 would be helpful.
Similar ask for Table 6 as for Table 5 (adding regions).
If I am reading the table correctly, it seems the heterogeneity is very high for Table 6. Consider performing a random effects meta-analysis or subgroup analysis. With this degree of heterogeneity, it seems to be more apples and orange comparison and the results are not very meaningful.
Interpretation of the results is less than ideal given the heterogeneity of the studies and further analysis is necessary, if a meta-analysis of these data are still considered important in this systematic review. The review alone might be more relevant.
Reviewer 2 Report
Chen et al present the results of their systemic review of published literature on the willingness of parents and guardians to vaccinate children or minors. Given the severity of the ongoing pandemic and the need to reach out to recalcitrant populations to increase vaccine uptake, the research addressed in this manuscript is critical. The authors provide a strong foundation for the design and implementation of future research by both analyzing current published literature and identifying gaps and under addressed research areas. The systematic review and meta-analysis are properly performed and the manuscript is well written and constructed. This paper is an ideal candidate for publication in Vaccines. Below are a few comments provided for improving the paper:
- I applaud the authors for including the search string used for the retrieval of publications in Pubmed in Table 1 but I would suggest moving the string itself to a supplementary figure and generating a more traditional table that summarizes the search terms used in the study. The current format makes it difficult to critically evaluate the search terms.
- The image quality in Figure 1 is poor causing the text within the figure to be fuzzy. The authors should improve the image quality of the figure before publication. Figure 2 has a similar issue although to a lesser extent. Efforts should be made to improve both.
- The authors rightly conclude in their manuscript that “given the limited quality and quantity of included studies, future studies with a rigorous design [are] necessary for the verification of our findings”, the authors should outline suggestions for the optimum design of future studies based on their analysis of the currently published literature in the discussion.
Round 2
Reviewer 1 Report
Therefore, new researches, especially longitudinal studies at different in-441 tervals, will be needed to investigate the actual vaccination behavior and attitudes when 442 vaccination programs for children commence.
The plural of research is research.
Author Response
Dear reviewer:
Thanks for your kind reminder. We have corrected the spelling mistakes (“researches” → “research”). We will be more careful in future work, preventing similar mistakes from happening again.